# A modified CD9 tag for efficient protein delivery via extracellular vesicles

Shojiro Inano[1,2,3]*, Toshiyuki Kitano[1]

**1** Department of Hematology, Kitano Hospital, Osaka, Japan, **2** Kansai Electric Power Medical Research Institute, Osaka, Japan, **3** Kyoto Innovation Center for Next Generation Clinical Trials and iPS Cell Therapy (Ki-CONNECT), Kyoto University Hospital, Kyoto, Japan

* shoin@kuhp.kyoto-u.ac.jp

## Abstract

Extracellular vesicles (EVs) are attracting growing attention for therapeutic use and as diagnostic markers, particularly for cancer. Although therapies based on small interfering RNAs are under intensive research, other therapeutic molecules, especially proteins, have not been sufficiently investigated. One of the major method for loading proteins into EVs is electroporation; however, it damages membrane integrity and requires repeated purification, precluding clinical applications. Thus, natural and efficient protein transfer is a prerequisite for the clinical application of protein-based EV therapy. Another prerequisite is an efficient endosomal escape, as most EVs incorporated into receptor cells result in endosomal degradation. Therefore, we generated a short CD9 (sCD9)-INF/TAT tag for efficiently transfers fused proteins to the EV and enhances endosomal escape to address the abovementioned problems. Interestingly, protein transfer via EVs drastically improved when the EV producer and receptor cells were cocultured, strongly indicating bystander effects of cells producing therapeutic proteins fused with a sCD9-INF/TAT tag. This method can be applied to a wide range of therapeutic technologies, including cellular transplantation or viral therapy.

## Introduction

Extracellular vesicles (EVs), including exosomes, are surrounded by a lipid bilayer membrane and secreted by most eukaryotic cells [1]. They are attracting growing attention as cancer prognostic and diagnostic markers, therapeutic targets, and anticancer drug carriers [2–5].

Focusing on the role of EV as a therapeutic tool, small interfering RNA (siRNA)-mediated gene silencing is the most widely investigated method. To introduce siRNAs into EVs derived from cultured cells, EV proteins fused with a peptide ligand can achieve siRNA transfer [6]. Another method for introducing siRNA into EVs is electroporation, and mutant KRAS-directed siRNA treatments, which have achieved excellent effects in pancreatic cancer mice models [7].

However, siRNA treatment does not compensate for missing factors or eliminate the slow turnover of proteins. In these cases, mRNA transfer via EVs becomes a promising method, and its gene expression in the brain of EV-treated or cell-transplanted mice has been reported

**Data Availability Statement:** Experimental files are available from Mendeley Data at DOI:10.17632/vx3ks9wsj9.1.

**Funding:** This work was partly supported by Japan Society for the Promotion of Science Award

Number: 19K16821. The funders had no role in study design, data collection and analysis, decision to publish, or preparation of the manuscript.

[8]. Nevertheless, although this technique is presently attracting attention because single mRNA can produce multiple therapeutic proteins, mRNA sizes are considered to hamper its efficient delivery via EVs.

Another approach is the direct transfer of therapeutic proteins into EVs, then their incorporation into receptor cells. However, electroporation should be omitted for clinical use because it potentially damages membrane integrity and EV stability *in vivo* [9]. Additionally, modifying therapeutic proteins to transfer to EVs naturally can spare multiple processes, including repeated EV purification. Nevertheless, two significant obstacles still exist: 1) how to efficiently transfer the protein of interest to the EV and 2) how to overcome endosomal degradation after EVs have been incorporated into receptor cells (endosomal escape). Here we started the development of universal tag-derived EV proteins, which simultaneously promote protein transfer into EV and endosomal escape, utilizing the HiBIT technology (Promega) to track the behavior or tagged proteins.

## Materials and methods

### Plasmid construction

The full list of plasmids is shown in S1 Table in S2 File. All plasmids were constructed using the Gibson assembly [10] or LR clonase II (Thermo Fisher). The vectors utilized for transient expression were derived from pCMV-GFP, a gift from Connie Cepko (Addgene plasmid # 11153).

### Cells

293T and HeLa cells were cultured in DMEM with low glucose (Wako) supplemented with 10% fetal bovine serum (FBS) and 100 U/mL penicillin/strep. To produce EVs for downstream assays, cells were cultured in advanced DMEM (Thermo Fisher) supplemented with 3% EV-depleted FBS.

### Transfection and generation of stable cell lines

Cells were seeded on the day before transfection. At 70% confluency, the cells were transfected with the indicated plasmids using Avalanche transfection reagent (EZ bioscience) according to the manufacturer's instructions. For the generation of stable cell lines, a DNA mixture containing a piggy bac vector and transposase (5:2 molar ratio) was used, followed by selection with 1 μg/ml puromycin or 10 μg/ml blasticidin 24 h after transfection.

### EV depletion from fetal calf serum

FBS (Gibco) was centrifuged at 100,000 g for 12 h (TLA-50 rotor in Optima-MAX-XP, Beckmann Coulter). Subsequently, the supernatant was processed with a 0.2-μM syringe filter (Polyether sulfone, Sartorius).

### EV isolation, nanotracking analysis, and quantification

EVs were isolated via ultrafiltration combined with size exclusion chromatography (SEC) [11,12] with a slight modification. Briefly, the culture medium was centrifuged at 4˚C ($200 \times g$ for 5 min followed by $3,000 \times g$ for 10 min and $10,000 \times g$ for 10 min), and the supernatant was processed through microfiltration (PES membrane, 0.2-μm, Sartorius). After concentration by ultrafiltration (Amicon Ultra-15, 100kDa, Merk) and washing with PBS thrice, the EVs were subjected to SEC (Smart SEC Mini EV Isolation System, SBI) according to the manufacturer's instructions. In some experiments, the SEC step was omitted. For EV collection by

ultracentrifugation, the concentrated EVs after ultrafiltration were centrifuged at $100,000 \times g$ for 60 min (TLS-55, Optima MAX-XP, Beckman Coulter), followed by washing with PBS twice. To validate the eluted EVs, analysis by NS 500 (Malvern Panalytical) was performed (a camera level of 8 and a measurement time of 30 s). For quantification, 10 μL of the EV suspension was measured by a BCA assay according to the manufacturer's instructions (Wako).

## Measurement of the HiBiT signals

The HiBiT tag is the smaller fragment of the split Nano Lucciferase, and the compensation of the larger fragment (LgBiT) enables the quantification of the HiBiT-tagged proteins. 293T cells were transfected with the indicated plasmids, followed by medium replacement after 4 h. The cells were cultured for another 36 h, and the culture supernatant was processed by the REUIS method as described above and was diluted to 1 ml with DMEM. Then, 50 μl of the EV suspension was transferred to a white 96-well plate, and EV HiBiT signals were measured by the NanoGlo HiBiT Lytic Detection System (Promega) according to the manufacturer's instructions. To measure the cellular HiBiT signal, the cells were collected by trypsinization and washed with PBS thrice. Subsequently, the cell pellets were resuspended in 2 ml of PBS and 50 μl was transferred to a white 96-well plate. The HiBiT measurement was conducted similarly as above. Each measurement was performed in triplicate. The transition rate (%) of the EV was calculated as follows:

$$\frac{\text{EV HiBiT signal} \times 10 \text{ (expected total signal)}}{\text{Cellular HiBiT signal} \times 20 \text{ (expected total signal)}} \times 100$$

## Encapsulation of saponin into EVs

We followed a previous report (Nakase and Futaki, 2015a) with scale modifications for this analysis. First, EVs (collected by ultrafiltration followed by size exclusion chromatography) were quantified by BCA assay. mixed with saponin (10 μg) in PBS (100 μL). Then, after electroporation, (poring pulse: two pulses (5 msec), transfer pulse: five pulses (20 V, 50 msec)) in a 1 cm electroporation cuvette at room temperature using a super electroplater NEPA21 TypeII (NEPA genes), removal of excess saponin was accomplished through ultrafiltration and washing using Vivaspin 500 (100 kDa, Sartorius). Concentrated EVs were finally added to the 293 T cells cultured on a 96-well plate, and cells were analyzed using a Cell titer glo (Promega) after 24 h of incubation.

## SLEEQ (Split Luciferase Endosomal Escape Quantification) assay

We principally followed a previous original report with minor changes [13]. First, 293T-LSA and HeLa-LSA cells were seeded at 20,000 cells per well into white 96-well microplates. The next day, purified EVs were added to the medium at the 75[th] dilution rate (15 ml medium concentrated to 200 μl through ultrafiltration). Cells were then incubated for 16 h (or an indicated time). Afterward, these cells were washed twice with PBS to remove excess EVs. Subsequently, a 25 μL diluted NanoGlo live cell substrate (Promega) and 75 μL fresh media were added to the cells to measure cytosolic signals. Then, luminescence measurements were made 10 min after substrate addition on Enspire (Perkin Elmer). Cells were later treated with 0.01% w/v digitonin after three wash rounds, after which they were incubated for 1 h at 37°C before substrate addition and luminescence measurement to measure total cellular associations.

For the SLEEQ assay under coculture conditions using Transwell, 100,000 EV producer cells (which express HiBiT-EGFP-sCD9 under a doxycycline-inducible promoter) were seeded on a Transwell insert (24-well scale, Griner). Next, the same number of 293 T-LHA or HeLa-LHA cells were seeded on a 24-well white plate (Porvair) and cultured separately for 24 h. Subsequently, the insert was placed in the well and cocultured for 48 h. This insert was finally removed, and the cells were washed with PBS three times. Eventually, 75 µL diluted NanoGlo live cell substrate (Promega) and 225 µL fresh media were added to the cells. The endosomal escape efficiency (%) was calculated as follows:

$$\frac{Cytosoli\ delivery}{Average\ of\ total\ cellular\ association} \times 100$$

For coculture in the same well, 10,000 EV producer cells and the same number of 293 T-LHA or HeLa-LHA were seeded on 96-well white plates. Then, after 48 h of incubation, the cells were washed with PBS three times, after which the SLEEQ assay was performed as described above.

## Immunoblotting and immunoprecipitation

Cells were washed twice with PBS and lysed in Radio Immunoprecipitation Assay buffer (50 mmol/L Tris-HCl Buffer (pH 7.6), 150 mM NaCl, 1% Nonidet P40 Substitute, 0.5% Sodium Deoxycholate, and 0.1% SDS) supplemented with a protease inhibitor cocktail, 25 unit/ml Benzonase (Millipore), 2 mM $MgCl_2$, and 1 mM $Na_3VO_4$ on ice for 30 min. Then, the lysates were briefly sonicated and centrifuged at 14000 x g for 10 min. The supernatant was analyzed by immunoblotting, as previously described [14]. For immunoprecipitation, 5 µg of purified EVs were treated with PBS or 0.5% triton/1% SDS/PBS for 2 h at RT and were then sonicated (Bio Ruptor) for 15 min. The antibodies used are listed in S2 Table in S2 File.

## Immunocytochemistry

Cells grown on a cover glass were washed with PBS twice and fixed in PBS containing 4% paraformaldehyde for 15 min. Then, the cells were permeabilized with 0.5% Triton X-100/PBS for 5 min. After blocking with 2% BSA/PBS, the samples were stained with antibodies diluted in 2% BSA/PBS for 1 h at RT, followed by washing with PBS thrice and immunostaining with secondary antibodies in 2% BSA/PBS for 1 h at RT (Alexa Fluor 488-conjugated anti-mouse IgG and Alexa Fluor 555-conjugated anti-rabbit IgG (Molecular Probes)). The antibodies used are listed in S2 Table in S2 File.

## Results

### Tetraspanins show efficient EV directivity and low levels of endosomal escape efficiency

For loading proteins into the EVs, generating chimeric proteins utilizing EV proteins is a classical method reported previously [15]. EV proteins include (i) membrane transport and fusion-related proteins, (ii) tetraspanins, including CD9, CD63, CD81, CD82, CD106, (iii) MVB-related proteins, including ALIX and TSG101, and (iv) other proteins, like Integrins [16–19].

Since HEK-293 T cells are currently the preferred parental cell lines for designer cell-based studies *in vivo* [17,18], we first analyzed the EV transition efficiency of candidate EV proteins, which can be fused with protein of interest in 293 T cells (S1A Fig in S1 File). From the immunoblotting-based quantification results, tetraspanins showed a relatively high EV directivity.

Next, 293 T cells were transfected with HiBiT-tagged EV proteins (Fig 1A), followed by cellular expression level and EV transition rate measurements. The HiBiT tag is the smaller fragment of the split Nano Lucciferase, and the compensation of the larger fragment (LgBiT) enables the quantification of the HiBiT-tagged proteins. Under transient expression, tetraspanins also showed a higher directivity to EVs (Fig 1B). Notably, even though previous reports have demonstrated that several tag sequences can induce EV directivity, the N-terminus acylation tag demonstrated the best efficiency [19,20]. It was also discovered that although the acylation tag exhibited a good EV transfer ability (around 10% of cellular proteins), it did not surpass that of CD9. Moreover, CD63 manipulation is difficult among tetraspanins due to massive glycosylation [21]. Therefore, we decided to manipulate CD9 afterward.

Subsequently, we assessed endosomal escape after cellular uptake using HiBiT-CD9. The SLEEQ assay was developed recently to quantify endosomal escape [13]. In this system, LgBiT, the more significant component of nanoluciferase, was anchored to an actin filament. After HiBiT-tagged proteins had been incorporated into receptor cells, HiBiT-tagged protein accessed LgBiT only when it successfully escaped from the endosome, enabling luminescence-based quantification of endosomal escape (S1B Fig in S1 File). Based on an original report where the internal tag was replaced with an HA tag, we generated the 293 T and HeLa cell line stably expressing LgBiT-HA-actin (293 T-LHA and HeLa-LHA) (S1C Fig in S1 File). Then, we introduced HiBiT-EGFP-CD9 into 293 T cells and purified EVs through serial centrifugation (200 × g, 3000 × g, 10,000 × g), microfiltration (0.2 μm) and ultrafiltration (100,000 Da), followed by SEC yielding particles with relatively uniform diameters (S1D Fig in S1 File). Afterward, concentrated EVs were added to 293 T-LHA and HeLa-LHA cells, followed by the SLEEQ assay. Results showed a drastic difference in the endosomal escape efficiency among cell lines that did not exceed 5% in both cell lines (Fig 1C), supporting the manipulation requirement that promotes endosomal escape.

Notably, when the SEC process was omitted, no significant change in particle diameter was observed (S1D Fig in S1 File); however, the recovery of EVs determined by the HiBiT signal was considerably reduced (S1E Fig in S1 File), resulting in reduced total cellular association and cytosolic HiBiT signal (Fig 1C). Therefore, we decided to isolate EVs by microfiltration and ultrafiltration as described above for three reasons: 1) in this study, our aim was to generate a new tag for protein delivery via EVs; hence, the removal of large apoptotic bodies or free proteins that were not encapsuled into the EVs was sufficient for these experiments; 2) the culture medium for the EV receptor cells included a large amount of serum proteins derived from FBS even after EV removal by ultracentrifugation; hence, we could not completely exclude the effect of serum proteins; 3) for *in vivo* use, the incorporation of the injected EVs was considered to be affected largely by serum proteins and other EVs; hence, the experiments completely devoid of contaminants were outside the scope of our current study.

Because CD9 is a pleiotropic protein that adversely affects the prognosis in patients with cancer [22,23], including intercellular communication in the immune system [24], it is possible that using the native form of CD9 as an EV tag can cause unexpected impacts *in vivo*. Therefore, we decided to manipulate CD9 to undermine its native function and the possibility of achieving better endosomal escape efficiency.

## Short CD9 exhibits better EV transition of proteins and endosomal escape efficiency than CD9

The tetraspanin proteins share a similar membrane topology with four membrane-spanning domains and the first and second extracellular loops, termed the short extracellular loop and large extracellular loop [25] (S2A Fig in S1 File). A previous study reported that the

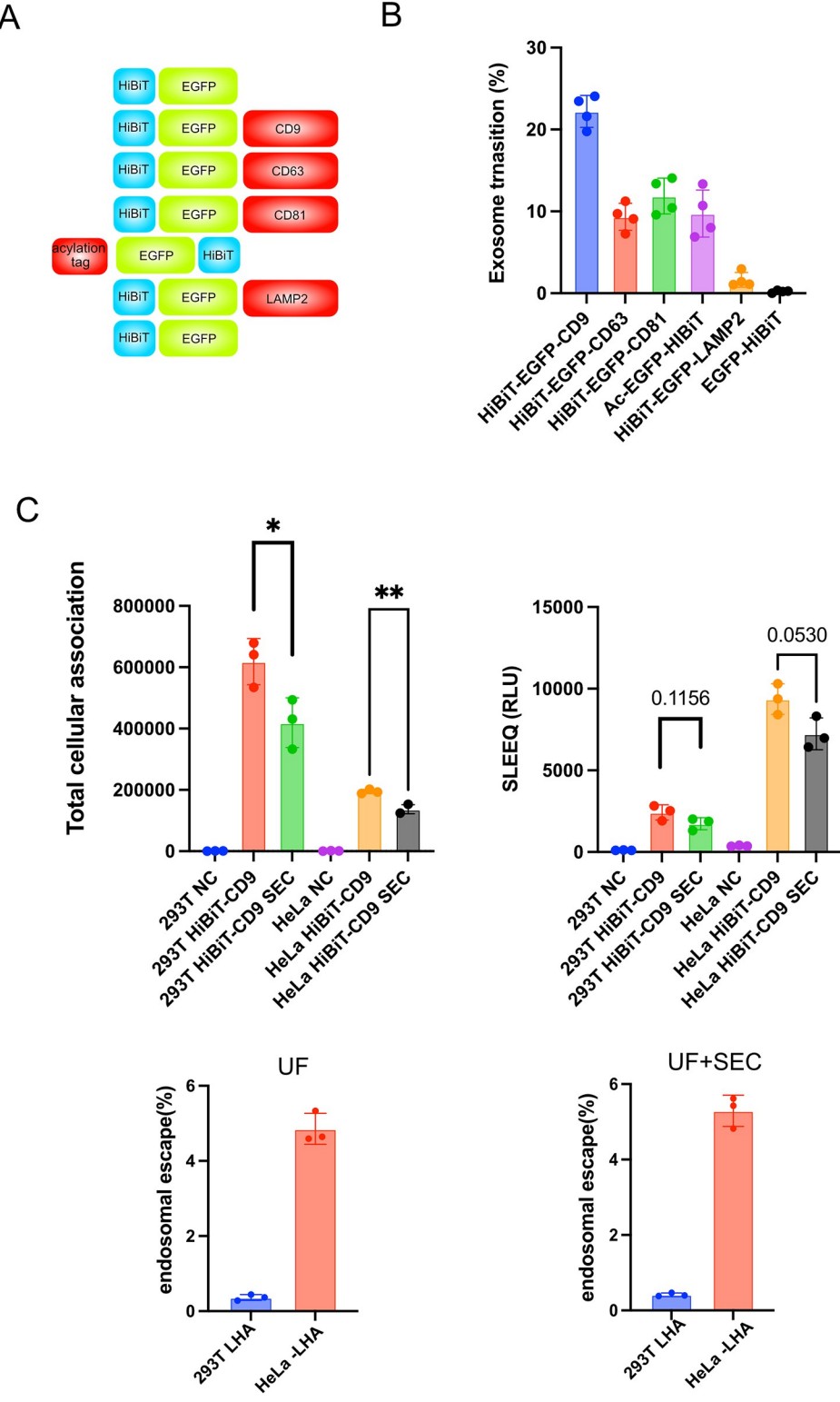

**Fig 1.** A) The construct of the plasmid in Fig 1B is illustrated. B) 293T cells were transfected with the indicated plasmids, and the HiBiT signal in EVs and the whole cell lysate were compared. The data are the averages (±SD) of three experiments. C) Purified EVs from 293T cells transfected with HiBiT-EGFP-CD9 were added to the indicated cells. After 12 h of incubation, total cellular association and cytosolic delivery were measured. The endosomal escape efficiency was calculated as follows: Cytosolic delivery/the average of total cellular association. In this experiment, the comparison of EVs purified via UF or UF-SEC was performed. The averages (±SD) of three experiments are displayed.

extracellular ends of short extracellular loop and large extracellular loop loosely packed with each other, creating a large central cavity inside the intramembranous region [25]. Given that the structure of the extracellular loop defines the intercellular functions of CD9, we first generated a short form of CD9 (sCD9) with truncated short extracellular loop and large extracellular loop (S2A Fig in S1 File). sCD9 revealed a similar distribution with CD9 (Fig 2A) and better transitions into EVs (Fig 2B). Importantly, when EVs containing EGFP-sCD9 were immunoprecipitated using a GST-GFP nanobody [26], the signal in immunoblotting turned positive only after the membranes were destroyed by triton X-100 treatment and sonication, indicating that EGFP was inside the EVs and sCD9 showed the same orientation as CD9 (S2B Fig in S1 File).

Next, we transfected 293 T cells with HiBiT-EGFP-CD9 and HiBiT-EGFP-sCD9, then analyzed their dynamics. sCD9 showed 1) increased cellular expression, 2) equivalent transition to EVs, 3) decreased incorporation, and 4) better endosomal escape efficiency than CD9 (Fig 2C). The same tendency was confirmed using HeLa-LHA as receptor cells (S2C Fig in S1 File). HeLa cells demonstrated superior endosomal escape efficiency compared to 293T cell line, but it may lead to the difficulty in the further improvement by sCD9 modification. That is why we decided to use 293T cell line in the subsequent experiments.

For the therapeutic molecules to function, cytosolic delivery is an essential process. Therefore, the absolute SLEEQ signal was the most critical parameter in this assay, indicating sCD9 as a better tag for EV transition of fused proteins than CD9.

## Insertion of a cell-penetrating peptide or fusogenic peptide into sCD9 affects endosomal escape

Although the endosomal escape efficiency of sCD9 was better than that of CD9, this efficiency was still low. Therefore, to improve the endosomal escape efficiency, we promoted membranous fusion by manipulating the extracellular lesion of sCD9. We focused on three types of peptides as candidate sequences for insertion: 1) cell-penetrating peptides, 2) fusogenic peptides, and 3) their combination.

Numerous studies have reported that although therapeutic peptides fused with Cell-penetrating peptides promote interactions among the plasma membrane and trigger endocytosis [27–29], improvement in endosomal escape might be limited [13]. Nevertheless, we hypothesized that cell-penetrating peptides exposed to EV membranes have EV-endosome membrane fusion, resulting in EV cargo release.

Alternatively, fusogenic peptides, mainly derived from viral membranous proteins, undergo topological changes in lower pH following endosome maturation to promote membrane fusion and endosomal escape [30,31]. Some papers also demonstrated the combination of cell-penetrating peptides and fusogenic peptides function synergistically for efficient cytosolic delivery [32].

Thus, we inserted the Cell-penetrating peptides/ fusogenic peptides or their combination in extracellular lesion of sCD9 (Fig 3A). First, we monitored the expression level and EV transition of these mutants. Although cell-penetrating peptides insertion decreased expression level and EV transition in general, fusogenic peptides had little impact on these parameters (Fig 3B, S3A Fig in S1 File). Importantly, total EV production measured by CD81-targeted ELISA did not show significant changes (S3B Fig in S1 File). Next, we measured the dynamics after cellular incorporation using the SLEEQ (Split Luciferase Endosomal Escape Quantification) assay, revealing that 1) Cell-penetrating peptides displayed on EVs did not enhance EV incorporation or endosomal escape and 2) while fusogenic peptides did not promote EV incorporation, they improved cytosolic delivery (Fig 3C and 3D). Importantly, the forced expression

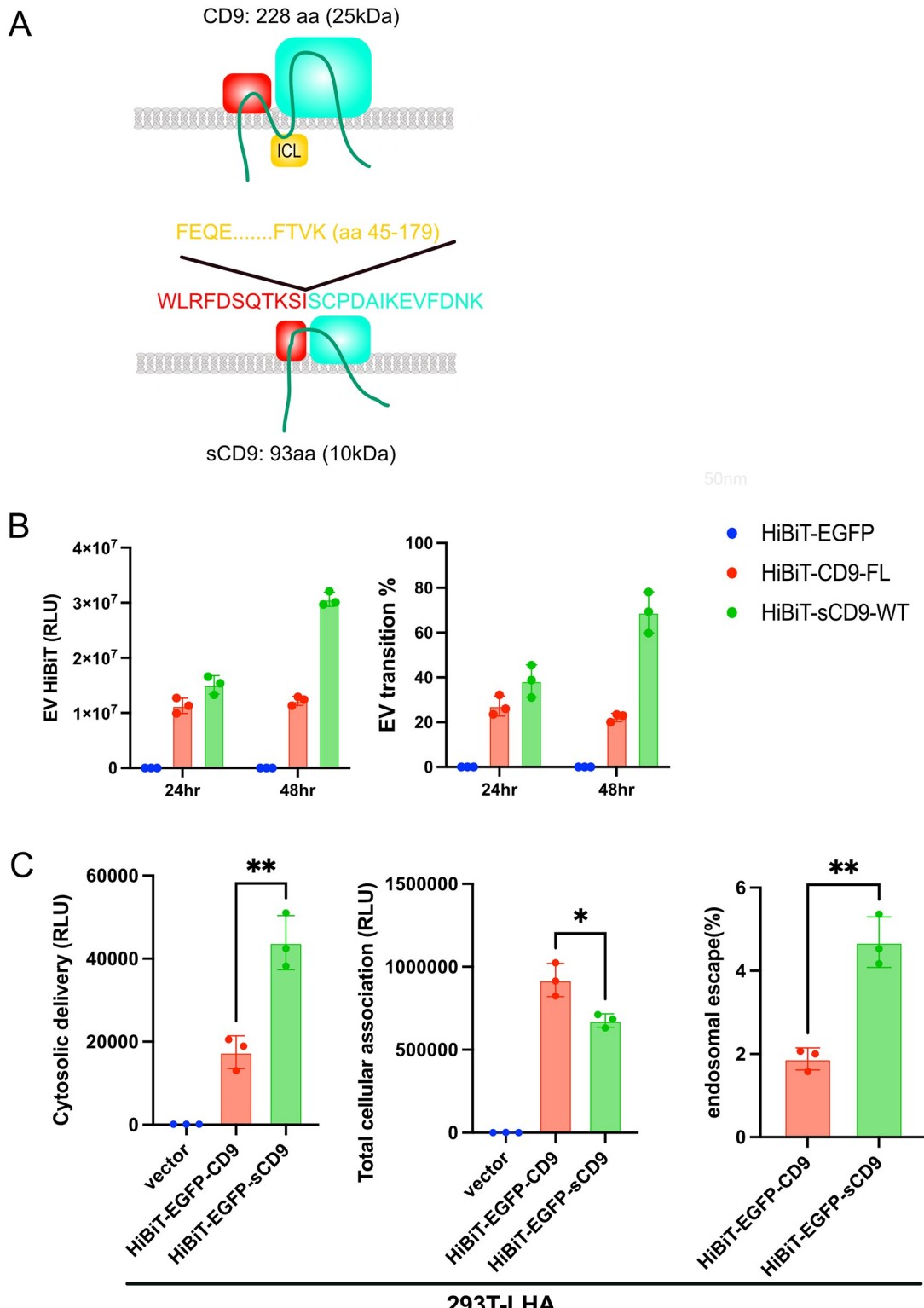

**Fig 2.** A) The structures of CD9 and short CD9 are illustrated. SEL; short extracellular loop, LEL; long extracellular loop, ICL; intracellular loop. B) 293T cells in 6-well plates were transfected with EGFP-CD9 or EGFP-sCD9. After 24 h of transfection, the culture medium was exchanged, and the cells were further incubated for the indicated time. EVs from the culture supernatant were purified and diluted to 2 ml (initial volume), and the HiBiT signals were measured. C) Purified EVs collected in B) were added to 293T-LHA cells. After 12 h of incubation, cytosolic delivery and total cellular association were measured.

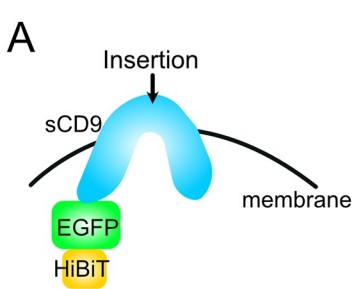

| Name | Sequence |
|------|----------|
| R9 | RRRRRRRRR |
| FR | FFLIPKGRRRRRRRRR |
| HL6 | CHHHHHHRRWQWRHHHHHHC |
| ZF5.3 | GWYSCNVCGKAFVLSRHLNRHLRHLRVHRRATAS |
| HL6 | CHHHHHHRRWQWRHHHHHHC |
| INF | CGLFEAIEGFIENGWEGMIDGWYGC |
| GALA | WEAALAEALAEALAEHLAEALAEALEALAA |
| INF-TAT | CGLFEAIEGFIENGLKGLIDAWYGYGRKKRRQRR |
| KL15-SN21 | KPVSLSYRCPCRFFESHVARAGGKLLKLLLKLLLKLLK |

**Fig 3.** A) The cell-penetrating peptides and fusogenic peptides inserted into sCD9 are shown. B) 293T cells in 6-well plates were transfected with the indicated plasmids in A), followed by medium exchange after 24 h. EVs were collected after another 48 h of incubation and diluted to 2 ml, and then the luminescence was measured to quantify the HiBiT-tagged protein level. The data are the averages (±SD) of three experiments. C–E) The EVs purified in B) were added to 293T-LHA cells. After 12 h of incubation, cytosolic delivery and total cellular association were measured. The endosomal escape efficiency was calculated as in Fig 1C. The data are the averages (±SD) of three experiments.

of these mutants did not also affect EV secretion dose, denying the effect on EV quantity (S3B Fig in S1 File). Hence, we focused on sCD9 -TAT with the highest SLEEQ signal and endosomal escape rate among the mutants above. Furthermore, since the INF/TAT peptide functions only after endosomal maturation and accompanies pH decrease [32], we tested the endosomal escape efficiency under the inhibition of a vacuolar proton pump with bafilomycin A1[33]. Expectedly, bafilomycin A1 treatment abrogated the effect of INF/TAT insertion (S3C Fig in S1 File).

Finally, we compared EVs collected by ultracentrifugation, which contained HiBi-T-EGFP-CD9, sCD9, sCD9-INF/TAT via immunoblotting. Again, no unexpected modification was observed (S3D Fig in S1 File).

## sCD9-INF/TAT promotes endosomal escape

Since the abovementioned experiments suggested that sCD9-INF/TAT facilitates endosomal escape, we assessed whether this chimeric protein achieved better cytosolic delivery of the EV cargo.

First, we generated 293 T cells stably expressing generated acyltag-fused EGFP-HiBiT (S4A Fig in S1 File), after which we introduced sCD9-WT or sCD9-INF/TAT under transient conditions. In this experiment, EVs contained two chimeric proteins (S4B Fig in S1 File). Next, EVs were collected via microfiltration, followed by ultrafiltration, and no significant HiBiT signal was observed among the samples (S4C Fig in S1 File). Subsequently, we added the collected EVs to 293 T-LHA and HeLa-LHA, confirming that sCD9-INF/TAT enhanced endosomal escape (Fig 4A and 4B). Notably, even in HeLa-LHA exhibiting relatively high endosomal escape efficiency (Fig 1C), the effect of INF/TAT insertion was significant.

Additionally, we introduced saponin, a potent protein synthesis inhibitor, with electroporation into EVs containing the sCD9 and sCD9-INF/TAT. Saponin functions leads to cell death only after endosomal escape, so that the cytotoxicity can be an indicator of endosomal escape. Saponin-containing EVs with sCD9-INF/TAT showed increased toxicity to 293 T cells (S4D Fig in S1 File), consistent with the results of SLEEQ assay (Fig 3D and 3E).

## EV-based protein transfer improves drastically during cocultures

Though EVs are promising tools for drug delivery, *in vivo* clearance is relatively rapid and requires repeated administration [34]. Nevertheless, cells stably secreting EVs that contain therapeutic molecules are potentially effective therapies in many fields of medication.

Based on these facts, we explored the possibility of cell transplantation therapy. First, we generated HiBiT-EGFP-sCD9 WT and INF/TAT under a doxycycline-inducible promoter, after which we cocultured with 293 T-LHA or HeLa-LHA through Transwell (0.4 μM) (S5A Fig in S1 File). Although the HiBiT signal was markedly different between the insert and the well (S5B Fig in S1 File), sCD9-INF/TAT revealed better cytosolic delivery under coculture conditions (Fig 5A and 5B). Interestingly, when those cells were cultured in the same well, the SLEEQ signal was much higher (Fig 5C and 5D). Results also showed that while EV producer cells were labeled with mCherry (under EGFP-sCD9 with IRES linkage), receptor cells showed only the EGFP signal (S5C Fig in S1 File), denying the possibility of mRNA transfer or cellular fusion.

For the mechanistic explanation of this discrepancy, we investigated the effect of EIPA, a potent inhibitor of macropinocytosis [35], using the SLEEQ assay. First, we added PKH67-stained EVs to 293T or HeLa cells and found that EIPA drastically reduces EV incorporation (S5D Fig in S1 File). Concanavalin A, an inhibitor of clathrin-mediated endocytosis [36], showed low impact, indicating that cells mainly depended on micropinocytosis for EV absorption, as previously described [37,38]. When the SLEEQ assay was performed under the

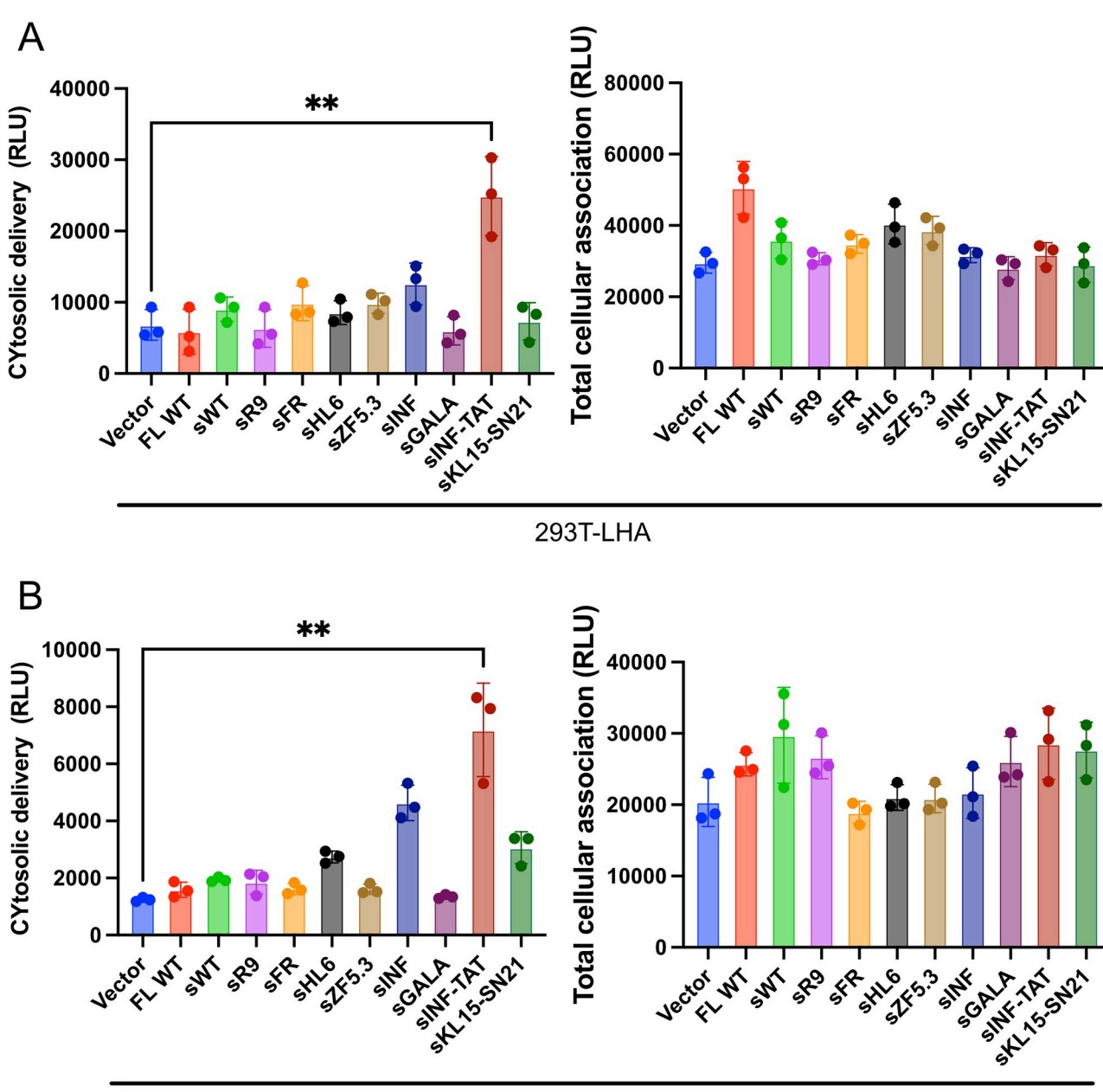

**Fig 4.** A–B) 293T cells stably expressing acylation-tagged EGFP were transiently transfected with the indicated plasmids (see S4B Fig in S1 File), and the extracellular vesicles were purified from culture medium as in Fig 3B and were added to 293T-LHA or HeLa-LHA cells. Cytosolic delivery and total cellular association were measured after 12 h of incubation.

same well coculture, EIPA considerably decreased the SLEEQ signal (Fig 5E and 5F), supporting efficient protein transfer via EVs among the neighboring cells.

## Discussion

In this report, we generated sCD9-INF/TAT derived from CD9. Subsequently, we observed that sCD9-INF/TAT promoted endosomal escape without decreasing expression levels and

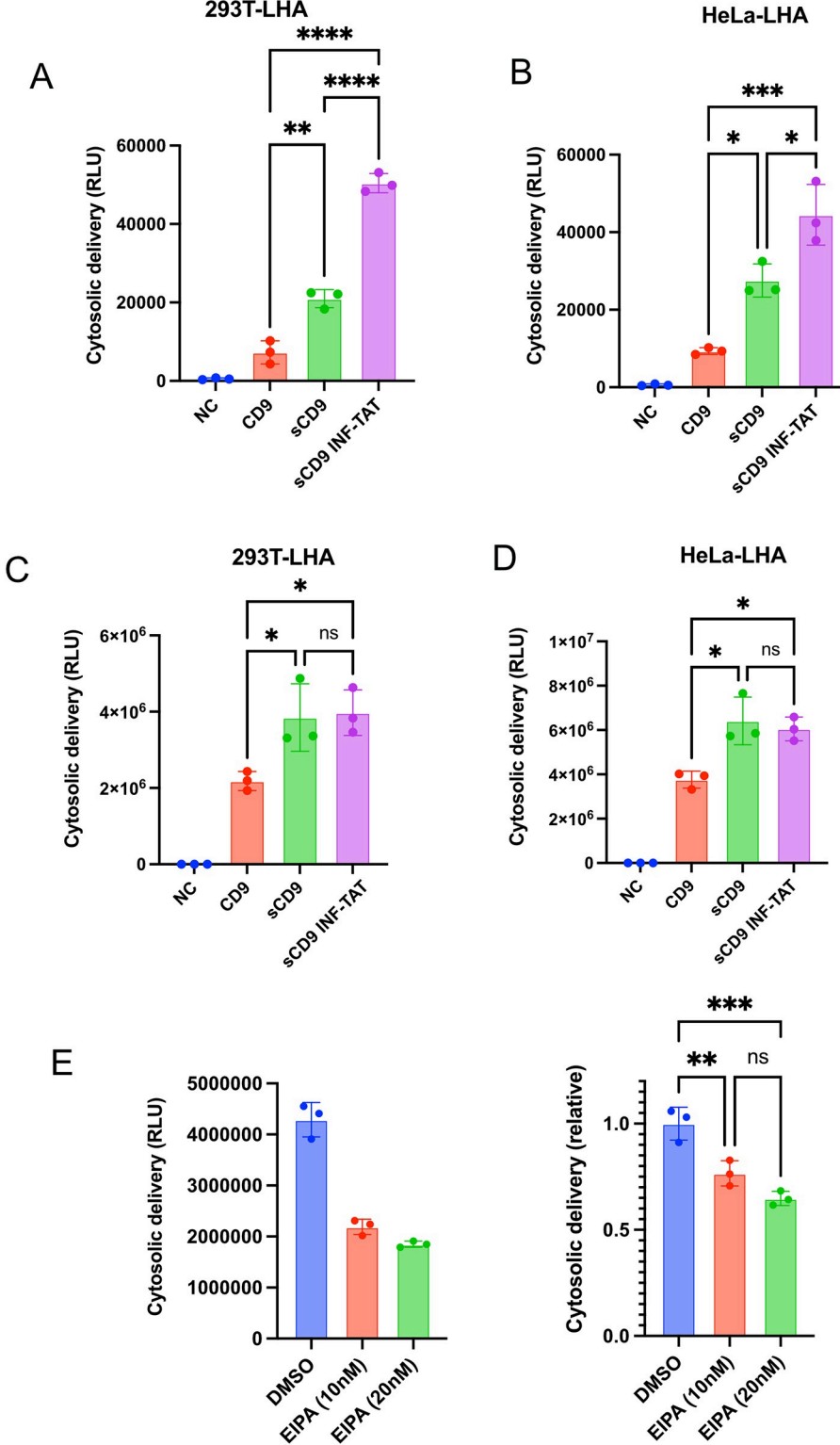

**Fig 5.** A–B) 293T cells expressing HiBiT-EGFP-sCD9 under a doxycycline-inducible promoter and 293T-LHA or HeLa-LHA cells were cocultured for 48 h via Transwell (S5A Fig in S1 File), and the cytosolic delivery in the receptor cells was measured by luminescence. C–D) Cells in A and B were cocultured in the same well for 24 h, and the cytosolic delivery was measured by luminescence. E–F) The experiment in C) was performed under the treatment of 10nM or 20nM of EIPA. For toxicity, EIPA was added 12 h after seeding (12-h treatment).

was feasible for the fusion protein. Therefore, our first purpose was to simultaneously achieve better incorporation and endosomal escape. Unfortunately, results showed that none of the sCD9 mutants promoted cellular incorporation.

Studies have shown endocytosis as a significant pathway for cellular uptake [39,40]. However, the targeting and uptake of s by recipient cells are poorly understood [41]. Since CD9 associates with a broad range of other functional proteins to exert cellular functions, CD9 over-expression may facilitate cellular uptake, and the truncation of and abrogates this effect. This study generated sCD9 that preserved the same orientation as CD9 via immunoprecipitation (S1B Fig in S1 File). Yet, it is unclear why the Cell-penetrating peptides displayed on the EVs failed to promote cellular uptake. One possible explanation is the density of Cell-penetrating peptides exhibited on the EVs, and because previous studies by Cell-penetrating peptides-fused proteins were conducted with much higher concentrations. Another explanation is the structure of the Cell-penetrating peptides displayed on the EVs. Though linked with sCD9 proteins with a relatively flexible linker sequence (GGGGS), insertion into the internal region of sCD9 may limit the function of Cell-penetrating peptides.

Focusing on the fusogenic peptides, we observed that sCD9-INF significantly enhanced endosomal escape and total SLEEQ signal (Fig 3D and 3E). HeLa-LHAAdditionally, the combination of INF and TAT with amino acid modifications synergistically improved endosomal escape, following another report [32]. However, these sequences still had room for optimization despite the limitation of structural flexibility that might prohibit the full functions of Cell-penetrating peptides/ fusogenic peptides.

Next, we showed that the continuous supply of proteins of interest via EVs from producer cells was effective in coculture experiments (Fig 5A and 5B). This experiment demonstrated the superiority of sCD9-INF/TAT as an EV tag that promotes endosomal escape. Surprisingly, when EV producer cells (293 T with HiBiT-EGFP-sCD9-INF/TAT) were cocultured with 293 T-LHA or HeLa-LHA cells in the same well, their cytosolic delivery increased drastically, but the superiority induced by INF/TAT insertion was not observed (Fig 5C and 5D). Given that the mRNA transfer was deniable (S5C Fig in S1 File), we can provide a rational explanation: 1) EV transfer occurs much more efficiently between neighboring cells, or 2) membranous proteins can be transferred to the adjoining cells independent of EVs. Therefore, to clarify this point, we performed a SLEEQ assay with EIPA, an endocytosis inhibitor, and a considerable decline in the SLEEQ signal supported the former hypothesis. However, the SLEEQ signal was not completely suppressed under EIPA treatment, which suggests the involvement of protein exchange mechanisms other than EVs. We lack further mechanistic insight accounting for this phenomenon, a future research objective.

In conclusion, we generated a single tag that simultaneously conferred efficient EV transfer and improved endosomal escape. For EV-based therapies, our method is simple and requires only a single EV purification process. Therefore, this process is proposed to significantly contribute to many research fields by sparing the complicated procedures required in EV preparation. Additionally, we observed that the EV-based protein transfer was affected mainly by the distance between the cells. Hence, our method is compatible with cellular transplantation therapy, virus-mediated protein expression, the spreading of therapeutic proteins to the surrounding cells.

## Supporting information

**S1 File. Supporting information corresponding to each main figure.**
(JPG)

**S2 File. List of plasmid vectors, antibodies.**
(XLSX)

## Acknowledgments

We thank Dr. Toshiyuki Kitano and colleagues in the hematology department for their great aid in medical practice. We also appreciate Dr. Makoto Mark Taketo and Dr. Huang Chenglong for continuously supporting this research.

## Author Contributions

**Conceptualization:** Shojiro Inano.

**Data curation:** Shojiro Inano.

**Formal analysis:** Shojiro Inano.

**Funding acquisition:** Shojiro Inano.

**Investigation:** Shojiro Inano.

**Methodology:** Shojiro Inano.

**Project administration:** Shojiro Inano.

**Resources:** Shojiro Inano.

**Software:** Shojiro Inano.

**Supervision:** Shojiro Inano, Toshiyuki Kitano.

**Validation:** Shojiro Inano.

**Visualization:** Shojiro Inano.

**Writing – original draft:** Shojiro Inano.

**Writing – review & editing:** Shojiro Inano.

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
