## [Decision Letter · Decision Letter 0]

9 Jul 2024

PONE-D-24-04765A Modified CD9 Tag for Efficient Protein Delivery via Extracellular VesiclesPLOS ONE

Dear Dr. Inano,

Thank you for submitting your manuscript to PLOS ONE. After careful consideration, we feel that it has merit but does not fully meet PLOS ONE’s publication criteria as it currently stands. Therefore, we invite you to submit a revised version of the manuscript that addresses the points raised during the review process.

We look forward to receiving your revised manuscript.

Kind regards,

Cheorl-Ho Kim, Ph.D.

Academic Editor

PLOS ONE

Journal Requirements:

"This work was partly supported by the Grants-in-Aid for Scientific Research from the Ministry of Education, Culture, Sports, Science, and Technology of Japan. The funders had no role in study design, data collection and analysis, decision to publish, or preparation of the manuscript."

**Additional Editor Comments:**

Dear Dr Inano,

Thank you for your submission to our PLOS One.

I have completed the first round review process, as I have attached the criticisms and merits.

As raised by our external experts, the manuscript can be further considered for publication.

I look forward to receiving your revised version and responses.

Thank you

Sincerely

Cheorl-Ho Kim phD Professor

Sungkyunkwan University,

Suwon Korea

Reviewers' comments:

Reviewer's Responses to Questions

**Comments to the Author**

1. Is the manuscript technically sound, and do the data support the conclusions?

Reviewer #1: Yes

Reviewer #2: Yes

Reviewer #3: Yes

2. Has the statistical analysis been performed appropriately and rigorously? 

Reviewer #1: Yes

Reviewer #2: Yes

Reviewer #3: Yes

3. Have the authors made all data underlying the findings in their manuscript fully available?

Reviewer #1: Yes

Reviewer #2: Yes

Reviewer #3: Yes

4. Is the manuscript presented in an intelligible fashion and written in standard English?

Reviewer #1: No

Reviewer #2: Yes

Reviewer #3: Yes

5. Review Comments to the Author

Reviewer #1: The authors performed serial protein engineering of CD9 to produce functional extracellular vesicles (EVs). They found that a truncated version of CD9, sCD9, that has better EV-sorting ability. Interestingly, the sCD9 offers better endosomal escape ability as well following cellular internalization. The claims are well substantiated by the data presented, but the language needs immense improvement for readership. Otherwise, I have a few minor points to bring up.

1. construct design in Fig s1a is very important, and should be placed in main figures for comprehension.

2. Fig 1a. Protein ladder is missing.

3. Fig 1b. The term "exosome transition" is rarely heard elsewhere. Try to use alternative terms, like EV-sorting efficiency?

4. Fig 1. The panels should be better organized. I mean, more artistic.

5. Fig 1c. Why Hela and HEK-293 cells demonstrate distinct endosomal escape profiles when treated with the same EVs?

6. All figures are without legends.

7. The abstract: "The current, high-efficient standard method for loading proteins into EVs is

16 electroporation," This is not true.

Reviewer #2: The authors demonstrated the results of a study in which they produced sCD9-INF/TAT tags and analyzed their effects for the purpose of natural and efficient protein delivery and efficient endosomal escape of EVs integrated into recipient cells. From a regenerative medicine perspective, the hurdles required to deliver proteins into EVs are extremely high, but the author's efforts to solve this problem by creating a specific protein structure are considered excellent. However, there are several key issues that need to be addressed from a research perspective.

1. A definition of sCD9 and application of the full name to the abbreviation (short CD9, sCD9) are required in the text. If possible given the content, it seems necessary for the author to insert the figure in Figure S2A into the main text.

2. In the main text, the authors reported that they created an sCD9 complex that enhances endosomal escape. Although ‘HeLa_LHA’ showed high results in the endosomal escape (%) data in Fig.1C, the author used ‘293T-LHA’ in subsequent experiments (Fig. 2). Why was the experiment conducted like that? Organize your justifications in the Results or Discussion section.

3. What is the proposal evidence for the CPP and FP presented in Figure 3A? Are there characteristics of each peptide? Or are there other effective peptides?

4. In the sentence of lane 263, it is concluded that forced expression of the mutation did not change EV particle diameter. However, Fig.S3C is only the peak diameter of the EV and does not represent the overall diameter distribution of the EV. It would be good to show the EV diameter for forced expression of each mutation.

5. From the data in Fig. S5B, the signals (RTU) of HibiT-sCD9 and HibiT-sCD9 INF-TAT appear to be almost similar, but the results in Figure 5 show a clear difference in cytosolic delivery. An explanation of Fig. S5B seems necessary.

6. It is necessary to provide the full name for abbreviations such as CPP or FP specified in the Manuscript. Furthermore, also to represent the concentration of used antibodies in the manuscript of supplemental table 2.

Reviewer #3: This paper addresses two aspects of employing extracellular vesicles (EVs) for the precise delivery of therapeutic molecules. It focuses on proteins rather than small interfering RNA since research on therapeutic peptides is somehow abandoned compared to that on siRNAs. One reason explaining the smaller popularity of therapeutic proteins is related to the fact that proteins are loaded into EVs by electroporation which damages membrane integrity and requires EVs’ repeated purification.

Hence, the authors propose a different way of protein transfer to EVs. Namely, they used an EV protein - tetraspanin CD9. To obtain a better transition into EVs they generated a short form of CD9 with a truncated short extracellular loop and a large extracellular loop, sCD9. sCD9 was used to generate an sCD9 INF-TAT tag that turned out to have the highest SLEEQ signal and endosomal escape rate among the studied mutants.

Thus, the authors proposed a method that allows loading a fused protein into EVs and facilitates its endosomal escape which should result in sufficient concentration of therapeutic peptide in cytoplasm.

This is an interesting and well-written paper. My only critical remark is related to the number of abbreviations that are not explained, e.g. HiBiT, SLEEQ, INF/TAT, etc. Please expand those abbreviations.

6. PLOS authors have the option to publish the peer review history of their article (what does this mean?). If published, this will include your full peer review and any attached files.

Reviewer #1: No

Reviewer #2: No

Reviewer #3: No

---

## [Author Response · Author response to Decision Letter 0]

6 Aug 2024

Reviewer #1: The authors performed serial protein engineering of CD9 to produce functional extracellular vesicles (EVs). They found that a truncated version of CD9, sCD9, that has better EV-sorting ability. Interestingly, the sCD9 offers better endosomal escape ability as well following cellular internalization. The claims are well substantiated by the data presented, but the language needs immense improvement for readership. Otherwise, I have a few minor points to bring up.

1. construct design in Fig s1a is very important, and should be placed in main figures for comprehension.

Thank you for your advice. We replaced Figure S1A and Figure 1A, and corresponding figure legends.

2. Fig 1a. Protein ladder is missing.

We added molecular weight marker.

3. Fig 1b. The term "exosome transition" is rarely heard elsewhere. Try to use alternative terms, like EV-sorting efficiency?

You are perfectly right. We corrected Fig 1b following your advice.

4. Fig 1. The panels should be better organized. I mean, more artistic.

Thank you. We modified the distribution of figure 1.

5. Fig 1c. Why Hela and HEK-293 cells demonstrate distinct endosomal escape profiles when treated with the same EVs?

Endosomal escape is a process that is both intricate and significant, and affected by the degradation ib endosomes. It is a natural occurrence that cells possess varying capacities for degrading endosomes and releasing them into the cytoplasm.

6. All figures are without legends.

We are sorry for the inconvenience. According to the instruction of the journal, figure legends are placed in the main text (Results). They are highlighted in red character in the “manuscript with track changes”.

7. The abstract: "The current, high-efficient standard method for loading proteins into EVs is

16 electroporation," This is not true.

Thank you. Appropriate changes have been made to the text in accordance with your recommendations . 

Reviewer #2: The authors demonstrated the results of a study in which they produced sCD9-INF/TAT tags and analyzed their effects for the purpose of natural and efficient protein delivery and efficient endosomal escape of EVs integrated into recipient cells. From a regenerative medicine perspective, the hurdles required to deliver proteins into EVs are extremely high, but the author's efforts to solve this problem by creating a specific protein structure are considered excellent. However, there are several key issues that need to be addressed from a research perspective.

1. A definition of sCD9 and application of the full name to the abbreviation (short CD9, sCD9) are required in the text. If possible given the content, it seems necessary for the author to insert the figure in Figure S2A into the main text.

Thank you. We added the abbreviation (short CD9) in the abstract and main text. In addition, we replaced Figure 2A and Figure S2A, and changed the corresponding figure legends.

2. In the main text, the authors reported that they created an sCD9 complex that enhances endosomal escape. Although ‘HeLa_LHA’ showed high results in the endosomal escape (%) data in Fig.1C, the author used ‘293T-LHA’ in subsequent experiments (Fig. 2). Why was the experiment conducted like that? Organize your justifications in the Results or Discussion section

Thank you for bringing that to my attention. In our experiments, HeLa cells demonstrated superior endosomal escape efficiency compared to 293T cell lines, but it may lead to the difficulty in the further improvement by sCD9 modification. That is why we decided to use 293T cell line in the series of experiments. We added the description above to the main text (Please see lines 237-240).

.3. What is the proposal evidence for the CPP and FP presented in Figure 3A? Are there characteristics of each peptide? Or are there other effective peptides?

The competition to develop constrained peptides (CPPs) and fragment peptides (FPs) is intense, and although some, such as R8, are frequently reported, others are only sporadically mentioned. Moreover, the role of these peptides when integrated into proteins, as demonstrated in this study, remains unclear. This collection of sequences was assembled through a manual search of the literature. Hence, it is possible that additional peptides may prove to be more effective.

4. In the sentence of lane 263, it is concluded that forced expression of the mutation did not change EV particle diameter. However, Fig.S3C is only the peak diameter of the EV and does not represent the overall diameter distribution of the EV. It would be good to show the EV diameter for forced expression of each mutation.

You are perfectly right. However, the figure would be too complicated if we put all the NTA (nanotracking analysis) data. Since it is not the fundamental data of this paper, we decided to omit Figure S3C and corresponding description.

5. From the data in Fig. S5B, the signals (RTU) of HibiT-sCD9 and HibiT-sCD9 INF-TAT appear to be almost similar, but the results in Figure 5 show a clear difference in cytosolic delivery. An explanation of Fig. S5B seems necessary.

It was surprising that the results of co-culture experiments were completely different from those by isolated EVs. We discussed this point in detail (Please see lines 379-389, in red character).

6. It is necessary to provide the full name for abbreviations such as CPP or FP specified in the Manuscript. Furthermore, also to represent the concentration of used antibodies in the manuscript of supplemental table 2.

Thank you. We replaced the CPP with cell-penetrating peptides, and FP with fusogenic peptides for the readability. In addition, we added the concentration of antibodies to table2.

Reviewer #3: This paper addresses two aspects of employing extracellular vesicles (EVs) for the precise delivery of therapeutic molecules. It focuses on proteins rather than small interfering RNA since research on therapeutic peptides is somehow abandoned compared to that on siRNAs. One reason explaining the smaller popularity of therapeutic proteins is related to the fact that proteins are loaded into EVs by electroporation which damages membrane integrity and requires EVs’ repeated purification.

Hence, the authors propose a different way of protein transfer to EVs. Namely, they used an EV protein - tetraspanin CD9. To obtain a better transition into EVs they generated a short form of CD9 with a truncated short extracellular loop and a large extracellular loop, sCD9. sCD9 was used to generate an sCD9 INF-TAT tag that turned out to have the highest SLEEQ signal and endosomal escape rate among the studied mutants.

Thus, the authors proposed a method that allows loading a fused protein into EVs and facilitates its endosomal escape which should result in sufficient concentration of therapeutic peptide in cytoplasm.

This is an interesting and well-written paper. My only critical remark is related to the number of abbreviations that are not explained, e.g. HiBiT, SLEEQ, INF/TAT, etc. Please expand those abbreviations.

Thank you very much for your peer review and positive feedback. We have tried to reduce abbreviations as much as possible, but we apologize for the still many abbreviations that impair the readability. HiBiT is a trademark of Promega, and INF and TAT are protein names, so they are difficult to explain. However, we have provided explanations for the remaining terms to the extent feasible.

---

## [Decision Letter · Decision Letter 1]

26 Aug 2024

A Modified CD9 Tag for Efficient Protein Delivery via Extracellular Vesicles

PONE-D-24-04765R1

Dear Dr. Inano,

We’re pleased to inform you that your manuscript has been judged scientifically suitable for publication and will be formally accepted for publication once it meets all outstanding technical requirements.

Kind regards,

Cheorl-Ho Kim, Ph.D.

Academic Editor

PLOS ONE

Additional Editor Comments (optional):

I am very much pleased to accept your revision.

Thank you for your patience in revision and waiting.

Thanks for your submission again.

Cheorl-Ho Kim

Editor

Reviewers' comments:

Reviewer's Responses to Questions

**Comments to the Author**

1. If the authors have adequately addressed your comments raised in a previous round of review and you feel that this manuscript is now acceptable for publication, you may indicate that here to bypass the “Comments to the Author” section, enter your conflict of interest statement in the “Confidential to Editor” section, and submit your "Accept" recommendation.

Reviewer #1: All comments have been addressed

Reviewer #2: All comments have been addressed

Reviewer #3: All comments have been addressed

2. Is the manuscript technically sound, and do the data support the conclusions?

Reviewer #1: Yes

Reviewer #2: Yes

Reviewer #3: (No Response)

3. Has the statistical analysis been performed appropriately and rigorously? 

Reviewer #1: Yes

Reviewer #2: Yes

Reviewer #3: Yes

4. Have the authors made all data underlying the findings in their manuscript fully available?

Reviewer #1: Yes

Reviewer #2: Yes

Reviewer #3: Yes

5. Is the manuscript presented in an intelligible fashion and written in standard English?

Reviewer #1: Yes

Reviewer #2: Yes

Reviewer #3: Yes

6. Review Comments to the Author

Reviewer #1: (No Response)

Reviewer #2: The author has attempted to answer all questions reasonably and to make corrections. This paper can now be published in the journal.

Reviewer #3: (No Response)

7. PLOS authors have the option to publish the peer review history of their article (what does this mean?). If published, this will include your full peer review and any attached files.

Reviewer #1: **Yes: **Wenyi Zheng

Reviewer #2: No

Reviewer #3: No

---

## [Editor Report · Acceptance letter]

8 Oct 2024

PONE-D-24-04765R1 

PLOS ONE

Dear Dr. Inano, 

I'm pleased to inform you that your manuscript has been deemed suitable for publication in PLOS ONE. Congratulations! Your manuscript is now being handed over to our production team.

Kind regards, 

on behalf of

Professor Cheorl-Ho Kim 

Academic Editor

PLOS ONE